# The distribution of registered occupational therapists, physiotherapists, and podiatrists in Australia

Engida Yisma[1]*, Vincent L. Versace[2], Martin Jones[1], Sandra Walsh[1], Sara Jones[1], Esther May[3], Lee San Puah[1], Marianne Gillam[1]

**1** Department of Rural Health, Allied Health & Human Performance, University of South Australia, Whyalla and Mount Barker, SA, Australia, **2** Deakin Rural Health, Faculty of Health, Deakin University, Warrnambool, VIC, Australia, **3** Clinical and Health Sciences, University of South Australia, Adelaide, SA, Australia

* engida.derbie@unisa.edu.au

## Abstract

### Background

In Australia, the distribution of occupational therapists, physiotherapists, and podiatrists density (per 10,000 population) by measure of location/rurality, usual resident population, and area-level socioeconomic status has not been described.

### Objective

To describe the national as well as states-and territories-wide distribution of registered allied health workforce—occupational therapists, physiotherapists, and podiatrists—by measures of rurality and area-level socioeconomic position in Australia.

### Methods

A linked data study that brings together (1) the location of health practitioners' principal place of practice from the Australian Health Practitioner Regulation Agency, (2) a measure of location/rurality—Modified Monash Model (MMM), and (3) an area-level measure of socioeconomic status—Index of Relative Socio-Economic Advantage and Disadvantage (IRSAD). The provider-to-population ratio (i.e., density) of three Australia's allied health workforce (occupational therapists, physiotherapists, and podiatrists) was calculated according to the MMM classifications (i.e., Modified Monash 1–7) and IRSAD quintiles at state and national level.

### Results

Nationwide, the density of occupational therapists and physiotherapists was highest in metropolitan areas (Modified Monash 1) and decreased with the increasing levels of the MMM categories. The national density of podiatrists was highest in Modified Monash 3 areas. The density of occupational therapists, physiotherapists, and podiatrists was highest in areas with IRSAD quintile 5 (i.e., the highest socioeconomic position) and decreased with the declining levels of the IRSAD quintiles nationwide. Moreover, there were notable disparities

**Data Availability Statement:** All the three data sources used in the study are publicly available. The data can be accessible online from the AHPRA (https://www.ahpra.gov.au/), ABS (https://www.

abs.gov.au/websitedbs/censushome.nsf/home/
seifa) and the Commonwealth Department of
Health (https://data.gov.au/data/dataset/modified-
monash-model-mmm-2019).

**Funding:** This study was funded by the
Commonwealth Department of Health via the Rural
Health Multidisciplinary Training (RHMT) Program.
The funder played no role in the design, conduct or
interpretation of the analyses.

**Competing interests:** The authors have declared
that no competing interests exist.

in the density of occupational therapists, physiotherapists, and podiatrists across each state
and territory in Australia when stratified by the MMM classifications and IRSAD quintiles.

## Conclusions

There was uneven distribution of registered occupational therapists, physiotherapists, and
podiatrists when stratified by measures of location/rurality and area-level socioeconomic
status across Australian jurisdictions. The density of these three groups of allied health
workforce tended to be more concentrated in metropolitan and most advantaged areas
while remote and most disadvantaged areas exhibited less allied health workforce distribu-
tion across each state and territory.

## Introduction

The health workforce plays a pivotal and indispensable role in shaping healthcare systems
worldwide. Furthermore, the healthcare workforce stands at the core of efforts to achieve the
United Nations Sustainable Development Goals and promote universal health coverage [1, 2].
The 2016 Global Strategy on Human Resources for Health: Workforce 2030 projected a global
shortfall of 18 million health workers by 2030 [3], encompassing both urban and rural areas.
The availability and accessibility of health workers continue to exhibit considerable disparities
within countries [2], owing to challenges in attracting and retaining health professionals in
both urban and rural regions. For instance, compared with Australians living in metropolitan
areas, residents in rural and remote areas experience poorer access to health care services, have
a higher prevalence of health risk factors, and have higher rates of injury, hospitalisation, and
death [4]. Similar health disparities exist between rural and metropolitan areas in other devel-
oped nations such as Canada and the United States of America (USA) [5]. Factors associated
with health inequalities include socioeconomic position (income, education, and occupation),
access to health care services, health workforce distribution, and occupational and physical
risks [6, 7]. These factors contribute to the wide variations in health inequalities across rural
and metropolitan areas.

Evidence suggests that the socioeconomic position of people in each area is an important
determinant of health outcomes in that area. For instance, evidence has shown that individuals
from lower socioeconomic groups experience higher rates of illness, disability, and mortality,
as well as shorter lifespans compared to those individuals from higher socioeconomic groups
[8]. Thus, improving access to material and social resources are important for addressing
health inequalities in society. Another key aspect in health inequality is health care service
delivery. Given that the health workforce is the main part of any health care system, the avail-
ability and accessibility of health workforce has a critical role in addressing the health inequali-
ties of people in different socioeconomic groups.

The allied health workforce, together with the medical practitioners, nurses, and midwives,
are the main part of Australia's health system that is intended to provide safe and affordable
health care for all residents [9]. There were about 133,400 registered allied health professionals
in Australia in 2018 [10]. The availability and accessibility of adequately skilled allied health
workforce such as occupational therapists, physiotherapists and podiatrists is essential to sup-
port people living with long-term physical health conditions [11]. On the other hand, a short-
age of allied health workforce can affect health service delivery with implications in the form of
adverse health outcome of the population, particularly in rural Australia where residents

experience higher levels of long-term physical and mental health conditions [11, 12]. Equitable distribution of allied health workforce across all areas is essential to alleviate health disparities.

There are few studies that described the distribution of the allied health workforce including, physiotherapists, and occupational therapists. For instance, Rodés *et al.* [13], in their 2021 study that utilised data from the Brazilian National Registry of Health Care Facilities, described the trends of the physiotherapy workforce-to-population ratio in Brazil and its regions. They found that the physiotherapy workforce-to-population ratio was variable according to care levels, and public and private sectors across all regions of Brazil [13]. Moreover, several studies conducted in Canada [14–16] found that the distribution of physiotherapists and self-reported physiotherapy use varies according to health regions and population (e.g., less physiotherapists in rural and remote areas), suggesting reduced access to physiotherapy services particularly in rural and remote areas. Furthermore, a study conducted by Lin and colleagues [17] in USA based on forecast models revealed that there is a shortage of occupational therapists in several states nationwide and the shortages are anticipated to rise in all states through 2030. Ned *et al.* [18] in their study that utilised data obtained from 'the Health Professions Council of South Africa' database, described the distribution and status of occupational therapists workforce. They found that there was uneven distribution of occupational therapists in South Africa, with the high proportions of occupational therapists were in urbanized provinces.

In Australia, uneven geographic distribution of health workforce, particularly the medical workforce, has been reported using different data sources. For instance, a study conducted by Joyce and Wolfe [19] using census data from 1996 and 2001 described the geographic distribution of the medical and non-medical primary health professions, including allied health professionals. The study reported that the supply of the general medical workforce in remote areas was lower when compared with metropolitan areas. Moreover, a 2021 study by Yisma *et al.* [20] using linked data from three data sources described the distribution of occupational therapists, physiotherapists, and podiatrists according to the Modified Monash Model and area-level socioeconomic position in South Australia. They found that the highest occupational therapists, physiotherapists, and podiatrists density (per 10,000 population) was in metropolitan areas and the lowest density of each of these health professionals was in very remote communities. However, the health workforce density, including allied health professionals density according to the socioeconomic characteristics of an area and geographic location has not been described at national level, or for other states and territories. Areas in Australia are ranked based on a composite measure of socioeconomic position such as Index of Relative Socio-Economic Advantage and Disadvantage (IRSAD). Versace *et al.* [21] conducted a national analysis of Australia accounting for area-level socio-economic conditions and population distribution, stratified by the Modified Monash Model to inform health workforce planning. They found that the majority of rural residents lived in areas with the lowest IRSAD categories. Describing the allied health workforce density by such socioeconomic position measures would be important to understand and address social inequalities and disadvantage. Moreover, given that both socioeconomic disadvantage and geographical remoteness are an important predictors of health outcome, [22] understanding the allied health workforce density by measure of socioeconomic position and geographical remoteness would be critical to inform the public health policy regarding allied health workforce recruitment and retention as well as management of allied health care service provision. Unlike previous studies, the objective of the present study was to integrate geographic factors (location/rurality) and area-level socioeconomic position to provide a new insight into the interplay between health workforce availability, socioeconomic factors, and geographic disparities. The purpose of the present study was to describe the national as well as states-and territories-wide distribution of

registered occupational therapists, physiotherapists, and podiatrists by measures of location/ rurality and area-level socioeconomic position in Australia.

## Materials and methods

### Data source

We linked data from three sources: (1) the public registration data of occupational therapists, physiotherapists, and podiatrists obtained from the Australian Health Practitioner Regulation Agency (AHPRA); (2) the Modified Monash Model (MMM) 2019 data from the Australian Department of Health; and (3) the Socio-Economic Indexes for Areas (SEIFA) data and usual resident population data obtained from the Australian Bureau of Statistics (ABS) based on the 2016 census. We combined these three data sources to explore the interplay between allied health workforce availability, measure of rurality, and socioeconomic position. This approach is important to deliver policy relevant insights aimed at promoting equitable healthcare access and outcomes. The SEIFA ranks areas in Australia according to relative socioeconomic advantage and disadvantage and consists of four indexes. These are the Index of Relative Socio-Economic Disadvantage (IRSD), the Index of Relative Socio-Economic Advantage and Disadvantage (IRSAD), the Index of Education and Occupation (IEO) and the Index of Economic Resources (IER). The AHPRA data were extracted and de-identified in April 2020. The MMM categories and the SEIFA index quintiles were combined with AHPRA data based on the practitioner's principal place of practice. The occupational therapists, physiotherapists, and podiatrists were included in this study because of limited funding to obtain the data from AHPRA and these professions are consistent with the University of South Australia's offerings at the time. Furthermore, these group of health professionals were chosen because they play a key role in the management of long-term physical health conditions such as diabetes, cardiovascular diseases, and arthritis in Australia [11].

### Study variables

The outcome measure of interest was the distribution of occupational therapists, physiotherapists, and podiatrists according to MMM and IRSAD. The MMM is a measure of location according to geographical remoteness and town/population size. The model quantifies remoteness and population size on a category scale of Modified Monash (MM) 1 to 7, which is based on the Australian Statistical Geography Standard—Remoteness Areas framework. For example, MM 1 indicates the location is a 'metropolitan area' and MM 7 indicates the locations that are 'very remote communities'. The IRSAD is one of the four SEIFA indexes generated by ABS and summarises information regarding the social and economic status of people and households within an area. The IRSAD index was chosen for the purpose of this study because it is a comprehensive index that combines several variables to measure both advantage and disadvantage across Australia. Consistent with the other SEIFA indexes, IRSAD scores are based on summary measures that represent an average of people and households in an area and they are not presumed to be applicable to all individuals within that area. A low IRSAD score (quintile 1) indicates relatively most disadvantage while a high IRSAD score (quintile 5) indicates relatively most advantage within an area. We categorized the IRSAD deciles into quintiles for the purpose of this study.

### Data analysis

We first calculated the proportion of occupational therapists, physiotherapists, and podiatrists by MMM categories and IRSAD quintiles for the whole of Australia as well as for each state

and territory (**S1 and S2 Tables**). Next, we summarized the total estimated usual resident population for Statistical Areas Level 1 (SA1) and Statistical Areas Level 2 (SA2) for whole of Australia as well as for each state and territory. SA1 and SA2 are part of the main structure of the Australian Statistical Geography Standard (ASGS), which classifies Australia into a hierarchy of statistical areas. The ASGS is developed to reflect the location of people and communities. We then calculated the provider-to-population ratio (density) of occupational therapists, physiotherapists, and podiatrists (i.e., per 10,000 population) stratified by the MMM categories and IRSAD quintiles for whole of Australia as well as for each state and territory. We defined the density of these group of allied health professionals as the total number of occupational therapists, physiotherapists, and podiatrists per 10,000 population, respectively by adopting definitions used by the World Health Organization's document [23]. To calculate the density of occupational therapists, physiotherapists, and podiatrists according to the MMM categories, we used SA1 data to align with the spatial unit used by the MMM. To calculate the density of occupational therapists, physiotherapists, and podiatrists according to IRSAD, we used SA2 data because according to ABS the SA2s are designed to represent communities that interact together socially and economically. All statistical analysis was conducted using Stata/SE version 17 (StataCorp., College Station, TX, USA).

## Ethics statement

Ethics approval and the need for participant consent for this study were exempted by the University of South Australia's Human Research Ethics Committee (Application ID: 203544).

## Results

### Density of occupational therapists by MMM categories and IRSAD

The density of occupational therapists within the MMM categories is presented in **Fig 1**. The national density of occupational therapists was highest in areas classified as MM 1 (10.9), and, except for MM 6, the density of occupational therapists decreased with the increasing levels of the MMM categories. Moreover, the density of occupational therapists varied according to each state and territory. For instance, in Victoria and New South Wales, the density of occupational therapists was highest in MM 2 areas, whereas in Queensland, South Australia, Western Australia and Australian Capital Territory, the density of occupational therapists was highest in MM 1 areas. In most states and territories, the pattern in the distribution of the occupational therapists density tended to be more concentrated in areas classified from MM 1–4, while areas classified from MM 5–7 exhibited less occupational therapists density. In the whole of Australia as well as in each state and territory that have areas classified from MM 5–7, the density of occupational therapists was high in MM 6 areas when compared to MM 5 and MM 7 areas.

**Fig 1** also shows that there were disparities in the density of occupational therapists among specific regions/areas across each state and territory in Australia. For instance, for MM2 areas, South Australia had the lowest density of occupational therapists (2.6) compared to other states and territories such as Victoria (12.6) and Northern Territory (11.4).

The density of occupational therapists according to IRSAD quintiles is presented in **Fig 2**. In the whole of Australia, the highest density of occupational therapists was found in areas with IRSAD quintile 5 (12.9) and decreased with the declining levels of IRSAD quintiles. However, the pattern in the distribution of occupational therapists density varied by IRSAD quintiles across each state and territory. For example, while Victoria, Queensland, and Western Australia had the highest density of occupational therapists in areas with IRSAD quintile 5,

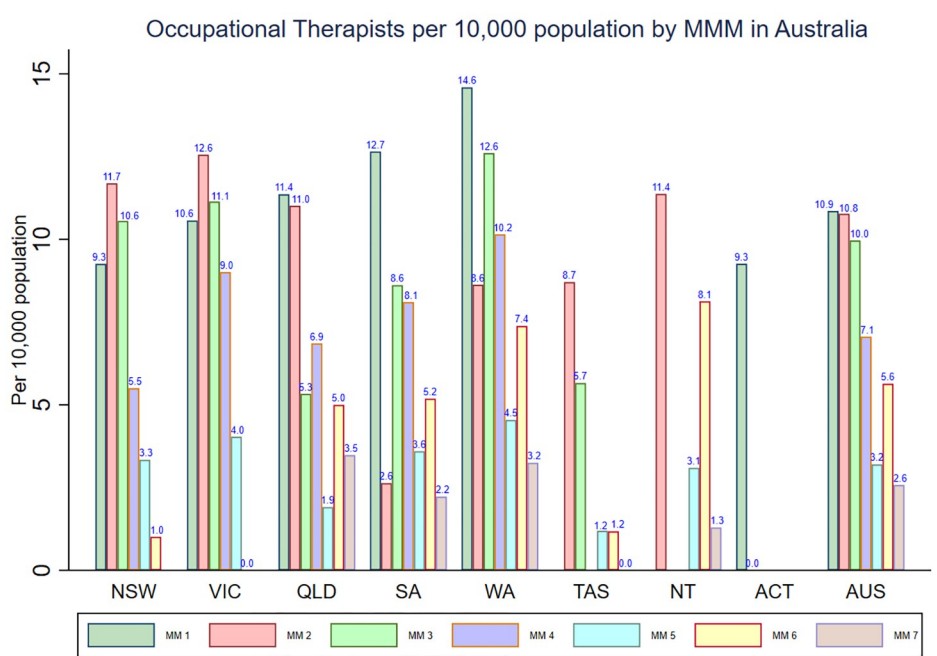

**Fig 1. Distribution of registered occupational therapists by MMM categories in each state and territory in Australia.**

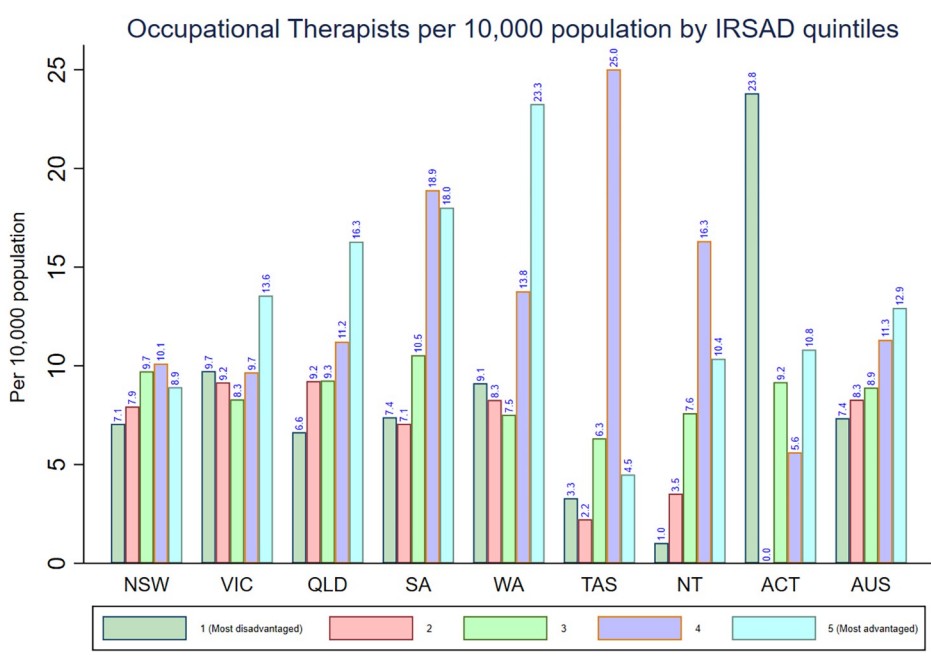

**Fig 2. Distribution of registered occupational therapists by IRSAD in each state and territory in Australia.**

New South Wales, South Australia, Tasmania, and Northern Territory had the highest density of occupational therapists in areas with IRSAD quintile 4.

## Density of physiotherapists by MMM categories and IRSAD

The national density of physiotherapists was highest in MM 1 areas (16.5), and, except for MM 6, the density of physiotherapists decreased with the increasing levels of the MMM categories. In the whole of Australia as well as in each state and territory that have areas classified from MM 5–7, the density of physiotherapists was high in MM 6 areas when compared to MM 5 and MM 7 areas. Northern Territory and Tasmania had the highest density of physiotherapists in MM 2 areas, while in all other states and territories, the density of physiotherapists was highest in areas classified as MM 1 (**Fig 3**).

**Fig 3** also shows that there were disparities in the density of physiotherapists among specific regions/areas across each state and territory in Australia. For instance, for MM2 areas, South Australia had the lowest density of physiotherapists (6.1) compared to other states such as Victoria (13.2) and Tasmania (13.3).

The physiotherapists density by IRSAD quintiles is presented in **Fig 4**. The highest national density of physiotherapists was found in areas with IRSAD quintile 5 (23.3) and decreased with the declining levels of IRSAD quintiles. The pattern in the distribution of physiotherapists density varied by IRSAD quintiles across each state and territory. For instance, South Australia, Tasmania, and the Northern Territory had the highest physiotherapists density in areas with IRSAD quintile 4 while the physiotherapists density in the other states was highest in areas with IRSAD quintile 5.

## Density of podiatrists by MMM categories and IRSAD

The podiatrists density by MM classification is presented in **Fig 5**. The national density of podiatrists was highest in MM 3 (2.7) areas, while it was lowest in MM 7 areas (0.6). Consistent

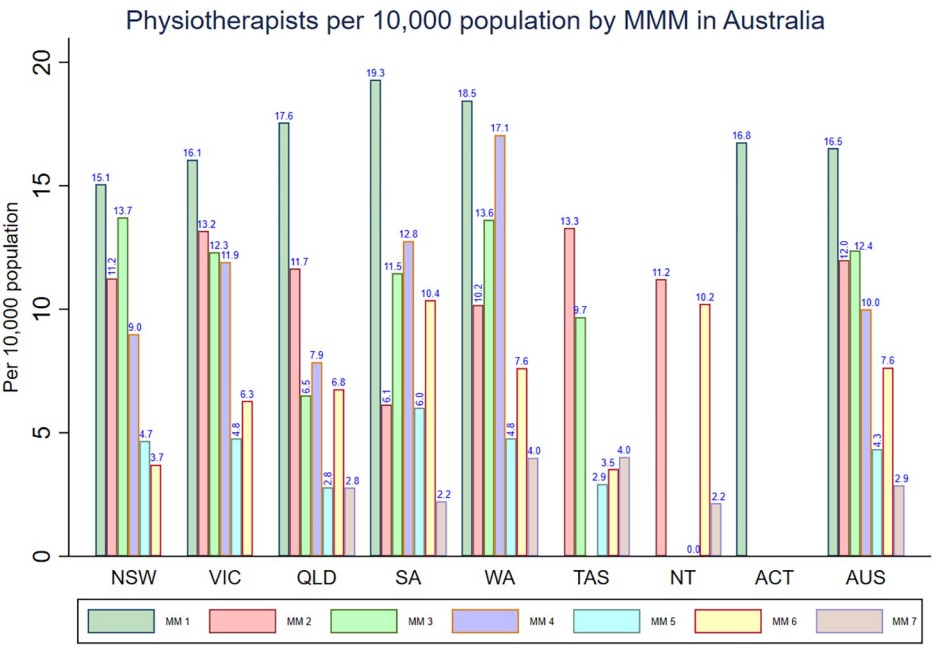

**Fig 3. Distribution of registered physiotherapists by MMM categories in each state and territory in Australia.**

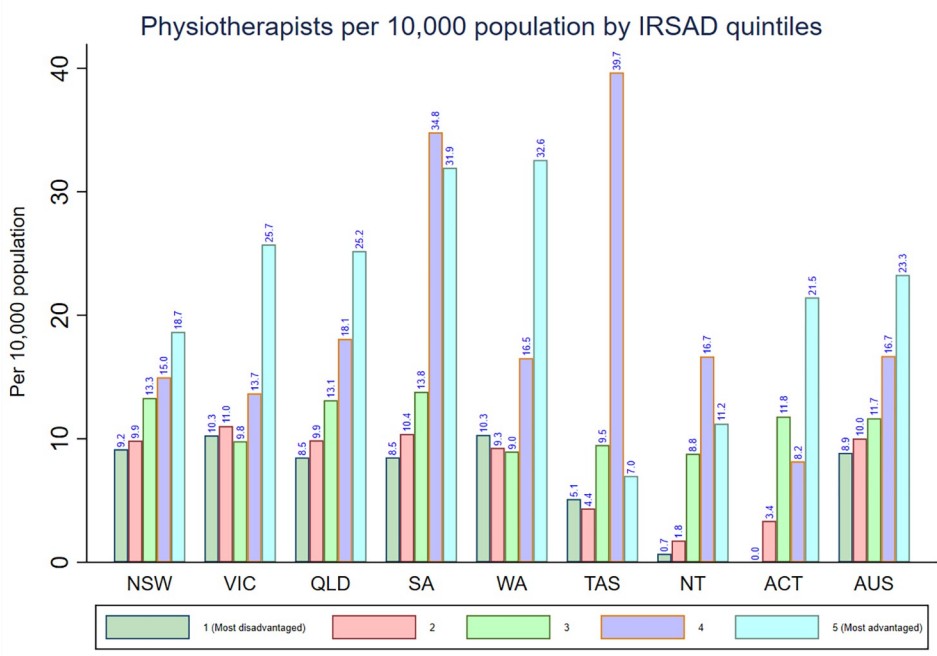

**Fig 4. Distribution of registered physiotherapists by IRSAD in each state and territory in Australia.**

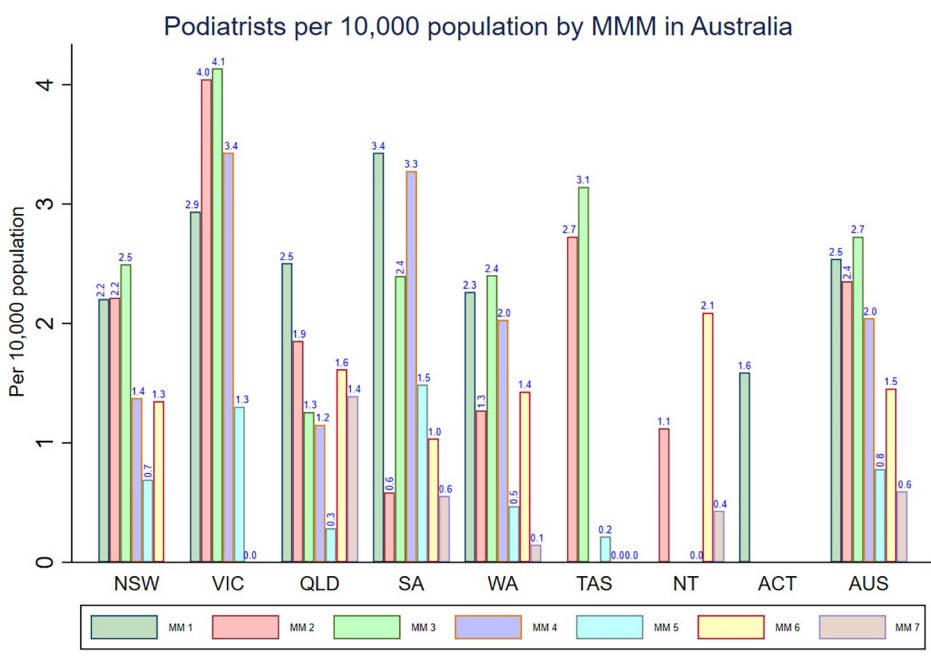

**Fig 5. Distribution of registered podiatrists by MMM categories in each state and territory in Australia.**

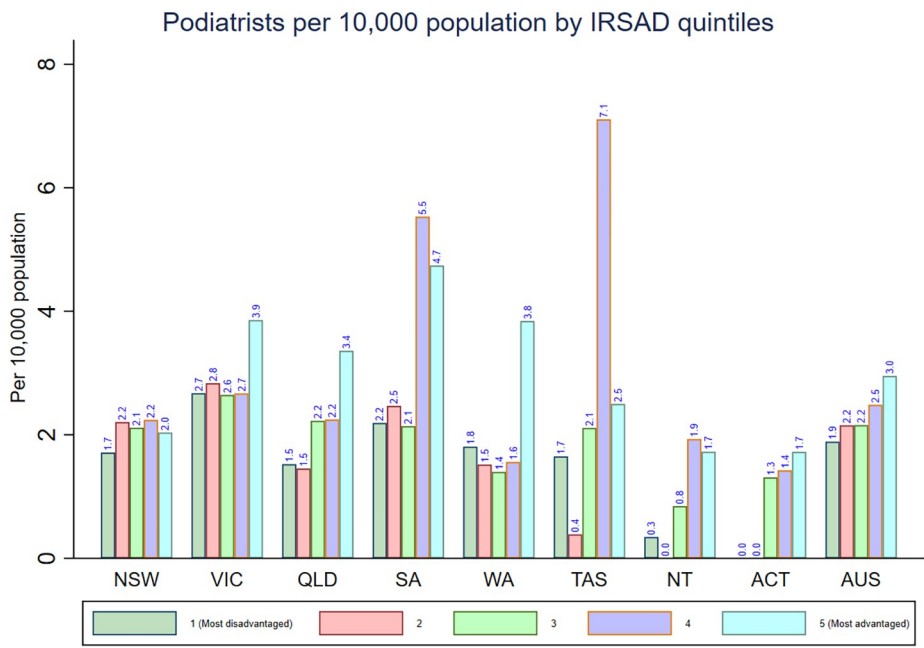

**Fig 6. Distribution of registered podiatrists by IRSAD in each state and territory in Australia.**

with the national pattern, New South Wales, Victoria, Western Australia, and Tasmania had the highest density of podiatrists in MM 3. However, Queensland, South Australia, and Australian Capital Territory had the highest density of podiatrists in MM 1 areas. Furthermore, the overall density of podiatrists tended to be more concentrated in areas classified as MM 1–4 across each state and territory while areas classified as MM 5–7 exhibited less podiatrists density.

**Fig 6** shows the density of podiatrists stratified by IRSAD quintiles. The highest national density of podiatrists was found in areas with IRSAD quintile 5 (3.0) and decreased with the declining levels of IRSAD quintiles. The pattern in the distribution of podiatrists density varied by IRSAD quintiles across each state and territory. For instance, New South Wales, South Australia, Tasmania, and the Northern Territory had the highest density of podiatrists in areas with IRSAD quintile 4 while the density of podiatrists in all other states was highest in areas with IRSAD quintile 5.

## Discussion

### Main findings

The present study revealed that there was uneven distribution of occupational therapists, physiotherapists, and podiatrists according to measures of rurality classification (MMM) and area-level socioeconomic status (IRSAD) across Australian jurisdictions. The national density of occupational therapists and physiotherapists was highest in areas classified as MM 1 and decreased with the increasing levels of the MMM categories. However, the national podiatrists density was variable across the categories of the MMM. In most states and territories, the pattern in the distribution of occupational therapists, physiotherapists and podiatrists density tended to be higher in areas classified from MM 1–4, while areas classified from MM 5–7

exhibited less occupational therapists density. Moreover, notable variations in the density of occupational therapists, physiotherapists, and podiatrists were also observed when analysing specific regions (such as MM 2 regions) in different Australian states and territories. In terms of IRSAD quintiles, the national density of occupational therapists, physiotherapists, and podiatrists was highest in areas with IRSAD quintile 5 and decreased with the declining levels of IRSAD quintiles. However, the pattern in the distribution of occupational therapists, physiotherapists, and podiatrists density varied by IRSAD quintiles across each state and territory. To the best of our knowledge, this is the first comprehensive study in Australia that has described the density of occupational therapists, physiotherapists, and podiatrists stratified by measures of rurality and area-level socioeconomic position. The findings of our study will provide a comprehensive information to inform allied health workforce planning in Australia.

Our finding that there was highest national density of occupational therapists and physiotherapists in 'metropolitan areas' (MM 1) and most advantaged areas (i.e., IRSAD quintile 5) may reflect better employment opportunities in metropolitan Australia. Conversely, the lower densities of occupational therapists and physiotherapists in 'very remote communities' (MM 7) and most disadvantaged areas (IRAD quintile 1) may explain the smaller range of employment and career opportunities in remote areas across Australia. A previous study conducted in South Africa by Ned *et al.* [18] found that there was spatial variation in the distribution of occupational therapists across jurisdictions. They found that the majority of occupational therapists were concentrated in urbanised provinces. Consistent with our findings, several studies conducted in Canada [14–16] found that the distribution of physiotherapists varies according to health regions and population size, suggesting reduced access to physiotherapy services, particularly in rural and remote areas as well as sparsely populated areas.

## Variations in the density of the three allied health workforce by a measure of rurality and jurisdiction

The highest states-wide density of occupational therapists, physiotherapists, and podiatrists varied according to MM categories across each state and territory in Australia. For instance, Victoria and New South Wales had the highest density of occupational therapists in MM 2 areas, whereas Northern Territory and Tasmania had the highest physiotherapists density in MM 2 areas. The highest density of the three allied health workforce in areas classified as MM 2 in Northern Territory and Tasmania is likely explained by absence of MM 1 areas in either jurisdiction. Moreover, in the whole of Australia as well as in each state and territory that have areas classified from MM 5–7, the density of physiotherapists was high in MM 6 areas when compared to MM 5 and MM 7 areas. Although it is unclear why MM 6 areas have higher density of these group of allied health practitioners, this finding is not surprising because the MMM scale is not meant to be in an ordinal scale and the health workforce density calculation considers total population of an area. Generally, the density of these three groups of allied health workforce tends to be more concentrated in areas classified from MM 1–4 across each state and territory while areas classified from MM 5–7 exhibited less health workforce density. As such, health workforce policies aimed to address workforce shortage need to focus on rural and remote communities. For instance, the Australian Rural Health Multidisciplinary Training (RHMT) program, which is one of several Commonwealth rural health workforce programs, aimed at recruiting and retaining health workforce has been in place for the past two decades. As it was emphasized in an independent review of the RHMT program conducted in 2020 [24], the program should continue investing in communities experiencing health workforce shortages. Consistent with the existing policy initiatives to address workforce maldistribution in high income countries [25], the RHMT program plays a key role to increase and

maintain the numbers of health workforce, including occupational therapists, physiotherapists, and podiatrists, thereby attempting to correct the geographic maldistributions. Moreover, funding and regulating allied health services by state and territory governments in their jurisdictions is essential to help correct uneven distribution of allied health workforce, particularly in rural and remote areas.

## Variations in the density of the three allied health workforce by area-level socioeconomic status and jurisdiction

The national density of occupational therapists, physiotherapists, and podiatrists was highest in IRSAD quintile 5 and decreased with the declining levels of the IRSAD quintiles. These figures suggest the presence of more allied health workforce in the most advantaged areas (IRSAD quintile 5) than the most disadvantaged areas (IRSAD quintile 1) in Australia. Similarly, in most states and territories, the occupational therapists, physiotherapists, and podiatrists density was highest in IRSAD quintiles 5. However, unlike the national distribution, South Australia, Tasmania, and Northern Territory had the highest density of occupational therapists, physiotherapists, and podiatrists in IRSAD quintile 4. The presence of higher density of these three allied health workforces in areas with IRSAD quintiles 4 or 5 is likely to be determined by metropolitan locations (i.e., MM 1). This is because metropolitan areas usually have higher IRSAD scores due to many households with high incomes and many people in skilled occupations. As metropolitan areas are classified as MM 1, the presence of high IRSAD scores in metropolitan areas demonstrates the inherent associations between the MMM classification and IRSAD scales. Moreover, our finding that the density of occupational therapists, physiotherapists, and podiatrists was lowest in areas of greatest socioeconomic disadvantage (i.e., IRSAD quintile 1) in Australia would appear to be intuitive because the most disadvantaged areas (quintile 1) tended to be in regional and rural areas, and it was previously reported that residents in rural and remote areas experience poorer access to health care services when compared with Australians living in metropolitan areas [4]. However, to properly analyse the overlap of remoteness and disadvantage, it is crucial to consider the specific context and relevant factors that could either intensify or alleviate the overlap between these two dimensions/measures.

## Strengths and limitations

The strengths of this study include the use of public registration data of occupational therapists, physiotherapists, and podiatrists from AHPRA, linked with publicly available national data from the Australian Department of Health and the Australian Bureau of Statistics. To link the data sets in the current study, we applied the methods described by Versace et al. [21] in their 2021 analysis of access, population distribution, and socio-economic status. A 2020 scoping review by Gillam et al. [26] recommended the use of linked data sets to help inform health workforce development and service planning in Australia. Moreover, Walsh et al. [25] in their 2020 study suggested that studies that focus on addressing rural allied health, nursing and dentistry workforce maldistribution should to be done at scale or with explicit links to coherent overarching policy. In the current study, the use of linked data from diverse sources enabled us to provide new insights into the distribution of registered occupational therapists, physiotherapists, and podiatrists across Australia, categorised by the MMM classifications and IRSAD quintiles. The study's findings, indicating a substantial distribution disparity of allied health workforce in Australia, with a concentration in metropolitan and advantaged areas and limited access in remote and disadvantaged regions, hold significant implications. From a clinical perspective, this underscores the critical need to customise healthcare delivery to address these

disparities, ensuring equitable access for all patients, regardless of their location or socioeconomic status. These insights also serve as valuable resources for researchers, informing evidence-based policies and interventions aimed at enhancing healthcare access and outcomes on a national scale. Furthermore, in addition to the national analysis of the distribution of occupational therapists, physiotherapists, and podiatrists by the MMM and IRSAD, we have also provided a detailed description of the distribution of these allied health professionals by the MMM classifications and IRSAD quintiles for each state and territory.

This study has some limitations. First, although the present study found a significant disparity in the allied health workforce distribution within each state and territory in Australia when classified by MMM and IRSAD quintile, further studies should explore these variations in detail across rural and disadvantaged areas. This is essential because some rural areas and disadvantaged populations may have unique characteristics that require targeted solutions. For instance, the recruitment of physiotherapists in certain rural regions may be easier than other rural areas, depending on contextual factors. Second, as the present study included only three allied health professionals, a future study needs to include a more exhaustive list of allied health workforce, noting that the AHPRA does not capture all allied health professions such as social workers. Third, as our study only focused on describing the distribution of three allied health workforce, further work is needed to understand factors associated with allied health workforce recruitment and retention in Australia. Finally, our analysis was based only on one of four SEIFA indexes (i.e., IRSAD). As each SEIFA index measure a different aspect of the socioeconomic conditions in each area, summarizing a different set of social and economic information, it would be important to study the distribution of allied health workforce by other SEIFA indexes in the future.

## Conclusions

This study found that there was uneven distribution of registered occupational therapists, physiotherapists, and podiatrists according to measures of rurality (MMM) and areal-level socioeconomic status (IRSAD) across Australian jurisdictions. The density of these three groups of allied health workforce tends to be more concentrated in metropolitan and the most advantaged areas (IRSAD quintiles 4 or 5) while remote and disadvantaged areas exhibited less allied health workforce density across each state and territory. The differing distribution of these group of allied health workforce suggests that a wide range of policy responses may be required to ensure equity of access to allied health care in all areas across states and territories. These measures are crucial in ensuring that healthcare services remain accessible and equitable for all Australians, irrespective of their geographic location or socioeconomic status.

## Supporting information

**S1 Table. Proportion of registered occupational therapists, physiotherapists, and podiatrists by MMM in Australia and in each state and territory, April 2020.**
(PDF)

**S2 Table. Proportion of registered occupational therapists, physiotherapists, and podiatrists by SA2 IRSAD quintiles in Australia, April 2020.**
(PDF)

## Acknowledgments

The authors would like to acknowledge the Commonwealth Department of Health, Rural Health Multidisciplinary Training (RHMT) Program.

## Author Contributions

**Conceptualization:** Engida Yisma, Vincent L. Versace, Martin Jones, Marianne Gillam.

**Data curation:** Vincent L. Versace, Martin Jones, Marianne Gillam.

**Formal analysis:** Engida Yisma.

**Methodology:** Engida Yisma, Vincent L. Versace, Martin Jones, Marianne Gillam.

**Resources:** Esther May.

**Software:** Engida Yisma.

**Supervision:** Vincent L. Versace, Martin Jones, Marianne Gillam.

**Visualization:** Engida Yisma.

**Writing – original draft:** Engida Yisma.

**Writing – review & editing:** Engida Yisma, Vincent L. Versace, Martin Jones, Sandra Walsh, Sara Jones, Esther May, Lee San Puah, Marianne Gillam.

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
