## [Decision Letter · Decision Letter 0]

13 Dec 2022

PONE-D-22-18539Geographic distribution of registered occupational therapists, physiotherapists and podiatrists in AustraliaPLOS ONE

Dear Dr. Yisma,

Thank you for submitting your manuscript to PLOS ONE. After careful consideration, we feel that it has merit but does not fully meet PLOS ONE’s publication criteria as it currently stands. Therefore, we invite you to submit a revised version of the manuscript that addresses the points raised during the review process.

We look forward to receiving your revised manuscript.

Kind regards,

Enamul Kabir

Academic Editor

PLOS ONE

Journal Requirements:

Reviewers' comments:

Reviewer's Responses to Questions

**Comments to the Author**

1. Is the manuscript technically sound, and do the data support the conclusions?

Reviewer #1: Partly

Reviewer #2: Yes

2. Has the statistical analysis been performed appropriately and rigorously? 

Reviewer #1: Yes

Reviewer #2: I Don't Know

3. Have the authors made all data underlying the findings in their manuscript fully available?

Reviewer #1: Yes

Reviewer #2: Yes

4. Is the manuscript presented in an intelligible fashion and written in standard English?

Reviewer #1: Yes

Reviewer #2: Yes

5. Review Comments to the Author

Reviewer #1: This is a relatively simple paper that describes the distribution of Australia’s allied health workforce (physio, OT, podiatry) at both a state and national scale, against 2 classifications measuring (1) rurality; (2) socio-economic status / IRSAD. There are many improvements required to its presentation.

Their title is misleading – they state “geographic distribution”, but IRSAD is technically not a geographical scale. It is a demographic classification – knowing that there are more services in a high IRSAD area does not by itself reveal geog distribution, without adding an extra geographic layer of where the high IRSAD areas are located.

I’m also not comfortable with their reference to the MM classification as a measure of “geographic access” – it is not. I am aware of reference #11 describing it in this way, but in this paper, it is actually quite confusing and misleading to use this phrase. The one ‘access’ measure this paper does utilise is the provider-to-population ratio (i.e. ‘density’), but confusingly the authors are not referring to that. The MM classification is a measure of ‘rurality’, it combines a geographic component (remoteness) with a demographic component (population size). The authors may be aware that the original measure of ‘remoteness’ (ARIA) had the word ‘accessibility’ in its name and some of the category labels used ‘accessible’; however this was highly criticised because it was NOT a measure of access, hence it was changed to the newer RA scale and variants of the word ‘access’ were wholly removed. The MM scale (which is partly defined by the RA scale) should also be applied this way.

I was somewhat disappointed by their presentation / discussion of the results. The authors almost wholly focus on the extreme categories only – those with the highest workforce density (can call this ‘access’) and those with the least. Unsurprisingly, this largely matches the extreme categories of both scales. The authors largely ignore whatever patterns are observed within the middle categories, which is arguably where much of the interest of these data is.

I think that part of the problem is that the authors have too many layers they’re trying to describe, forgetting that ‘distribution’ is the primary interest. Having 3x clinician types means that there are three patterns to describe, with little gained from if they focused on one. Jurisdictional differences may be important to some service planners, but generally add little to the paper’s value. Moreover, having 2x distribution scales doubles the observed patterns. Specific distributional differences should be the majority point of focus (e.g. what is the step down/up ratio of density (access) between categories of the MM / IRSAD classifications? Why is MM-6 generally ‘better’ than MM-5 / MM-5? (PS: this is not surprising, the MM scale is not meant to always be ordinal).

Where required, the authors don’t clarify certain patterns – chief example is the highest density in NT and Tas being in MM-2, rather than MM-1 (which is 100% explained by there being no MM-1 locations in either jurisdiction). Also, the authors at no point discuss the inherent association(s) between the MM and IRSAD scales. IRSAD differences (particularly at the higher end) will likely mostly be determined by metropolitan locations (ie. within MM-1). The last sentence prior to the Strengths directly assumes assoc between lower density and lower IRSAD relates to rural communities, but they have never actually demonstrated a linkage between MM and IRSAD.

Reflecting on my earlier point, the first sentence of the conclusion is nonsensical – ‘access’ belongs with the uneven distribution part of the sentence. The next sentence is not appropriate (‘accessible’ and ‘least accessible’ are incorrect labels). The last sentence is OK, with ‘equity of access’ being about the service levels (density). The Abstract conclusion needs to be similarly fixed up.

Another nonsensical statement is at lines 257-259 (“Conversely, the lower densities…”), which essentially is saying the ‘lower access (density)’…may explain the ‘lower access (availability)’! Following this, the heading “Variation by access and jurisdiction” needs changing.

One paragraph didn’t make sense to me – starting at line 189; it twice refers to NSW, SA, Tas, NT, ACT but I can’t determine why they are stated twice.

Reviewer #2: Major comments:

1. While this may be the first study of this kind in Australia, this type of study has been done in other countries including USA and Canada. The authors are encouraged to expand introductions and discussions comparing their findings with other studies. Are there any novel findings from this particular study?

Minor comments:

2. Abstract: The authors are requested to include the study objective after the background.

3. The background needs to describe in more detail why this study is so critical?

4. Discussion section also did not present according to the research purpose and result. The authors are encouraged to Interpret and explain study results, and critically compare to previous studies. For example, at first occupational therapist then physiotherapist and finally podiatrist.

5. Page 17, line 316-317. This sentence is not a limitation as the authors are focusing their objective based on three professions- occupational therapist, physiotherapist and podiatrist. Please drop it.

6. PLOS authors have the option to publish the peer review history of their article (what does this mean?). If published, this will include your full peer review and any attached files.

Reviewer #1: No

Reviewer #2: **Yes: **Mohammad Habibur Rahman Sarker

---

## [Author Response · Author response to Decision Letter 0]

1 Feb 2023

"Geographic distribution of registered occupational therapists, physiotherapists and podiatrists in Australia" (MS: PONE-D-22-18539). 

Dear Editor,

We are very grateful for the constructive comments from the editor and reviewers. We have taken due consideration of the comments provided and made the necessary revisions to the manuscript. All page and line numbers refer to the marked copy of the manuscript.

Response to Editorial comments

COMMENT: Please ensure that your manuscript meets PLOS ONE's style requirements, including those for file naming. 

AUTHORS’ RESPONSE: In the revision, we have re-formatted our manuscript according to the PLOS ONE’s style, including the file naming. 

COMMENT: Please provide additional details regarding participant consent. In the ethics statement in the Methods and online submission information, please ensure that you have specified (1) whether consent was informed and (2) what type you obtained (for instance, written or verbal, and if verbal, how it was documented and witnessed). If your study included minors, state whether you obtained consent from parents or guardians. If the need for consent was waived by the ethics committee, please include this information.

AUTHORS’ RESPONSE: Given that we have utilised publicly available data sources, ethics approval and the need for participant consent for this study were exempted by the University of South Australia’s Human Research Ethics Committee (Application ID: 203544). In the revision, we have stated this in the methods section of our manuscript, and we have added the same text to the “Ethics Statement” field of the submission form. Our revision can be found in the methods section track changed. 

Response to Reviewer #1

COMMENT: This is a relatively simple paper that describes the distribution of Australia’s allied health workforce (physio, OT, podiatry) at both a state and national scale, against 2 classifications measuring (1) rurality; (2) socio-economic status / IRSAD. There are many improvements required to its presentation.

AUTHORS’ RESPONSE: In the revision, we have significantly improved the presentation of results and interpretation based on the reviewers’ comments. Our revision can be accessed in the track changes in the revised manuscript. 

COMMENT: Their title is misleading – they state, “geographic distribution”, but IRSAD is technically not a geographical scale. It is a demographic classification – knowing that there are more services in a high IRSAD area does not by itself reveal geog distribution, without adding an extra geographic layer of where the high IRSAD areas are located.

AUTHORS’ RESPONSE: We apologise for our misleading title. We have removed the term ‘geographic’ from our title and have revised the title as “The distribution of registered occupational therapists, physiotherapists and podiatrists in Australia". We appreciate that although IRSAD measures the socioeconomic status of an area, it is a sociodemographic measure and may not be used to describe geographic classification on its own. We have revised the text of our manuscript accordingly and our revision can be found in track changes in the revised manuscript.

COMMENT: I’m also not comfortable with their reference to the MM classification as a measure of “geographic access” – it is not. I am aware of reference #11 describing it in this way, but in this paper, it is actually quite confusing and misleading to use this phrase. The one ‘access’ measure this paper does utilise is the provider-to-population ratio (i.e., ‘density’), but confusingly the authors are not referring to that. The MM classification is a measure of ‘rurality’, it combines a geographic component (remoteness) with a demographic component (population size). The authors may be aware that the original measure of ‘remoteness’ (ARIA) had the word ‘accessibility’ in its name and some of the category labels used ‘accessible’; however, this was highly criticised because it was NOT a measure of access, hence it was changed to the newer RA scale and variants of the word ‘access’ were wholly removed. The MM scale (which is partly defined by the RA scale) should also be applied this way. 

AUTHORS’ RESPONSE: Thank you for the comments. We agree that the MMM classification should not be used as a measure of “geographic access”. We have removed the phrase “geographic access” and replaced it with the appropriate phrase throughout the manuscript. We have also focused on the ‘Provider-to-population ratio (i.e., ‘density’)’ as a measure of access as per your suggestion in the revised manuscript. Our revision can be found as track changes in the revised manuscript. 

COMMENT: I was somewhat disappointed by their presentation / discussion of the results. The authors almost wholly focus on the extreme categories only – those with the highest workforce density (can call this ‘access’) and those with the least. Unsurprisingly, this largely matches the extreme categories of both scales. The authors largely ignore whatever patterns are observed within the middle categories, which is arguably where much of the interest of these data is.

AUTHORS’ RESPONSE: Thank you for this comment. In the revision, we have considered all categories of the MMM and IRSAD classification when presenting and discussing the results in our revised manuscript. In the whole of Australia, the highest density of the three allied health workforce was found in areas classified as MM 1 and decreased with the increasing levels of the MMM categories. Furthermore, in the whole of Australia, the highest density of the three allied health workforce was found in areas with IRSAD quintile 5 and decreased with the declining levels of IRSAD quintiles. However, the pattern in the distribution of the three allied health workforce densities varied widely by the MMM categories and IRSAD quintiles across each state and territory. Our revision can be accessed in track changes in the “Results” and “Discussion” sections. 

COMMENT: I think that part of the problem is that the authors have too many layers they’re trying to describe, forgetting that ‘distribution’ is the primary interest. Having 3x clinician types means that there are three patterns to describe, with little gained from if they focused on one. Jurisdictional differences may be important to some service planners, but generally add little to the paper’s value. Moreover, having 2x distribution scales doubles the observed patterns. Specific distributional differences should be the majority point of focus (e.g. what is the step down/up ratio of density (access) between categories of the MM / IRSAD classifications? Why is MM-6 generally ‘better’ than MM-5 / MM-5? (PS: this is not surprising; the MM scale is not meant to always be ordinal).

AUTHORS’ RESPONSE: In the revision, we have focused on describing the distributional differences of the three allied health workforce densities stratified by the MMM and IRSAD in the whole of Australia as well as in each state and territory as per your comments. Our revision can be found in track changes in the text of the revised manuscript. 

COMMENT: Where required, the authors don’t clarify certain patterns – chief example is the highest density in NT and Tas being in MM-2, rather than MM-1 (which is 100% explained by there being no MM-1 locations in either jurisdiction). Also, the authors at no point discuss the inherent association(s) between the MM and IRSAD scales. IRSAD differences (particularly at the higher end) will likely mostly be determined by metropolitan locations (ie. within MM-1). The last sentence prior to the Strengths directly assumes assoc between lower density and lower IRSAD relates to rural communities, but they have never actually demonstrated a linkage between MM and IRSAD.

AUTHORS’ RESPONSE: Thank you for this comment. In the revision, we have added sentences to the revised manuscript to explain the distributional patterns observed in the Northern Territory and Tasmania as per your suggestion (see page 19, lines 334-336). We have also added a couple of new sentences to the revised manuscript to explain the inherent association between the MMM and IRSAD classifications in the revised manuscript (see page 21, lines, 368-373). 

COMMENT: Reflecting on my earlier point, the first sentence of the conclusion is nonsensical – ‘access’ belongs with the uneven distribution part of the sentence. The next sentence is not appropriate (‘accessible’ and ‘least accessible’ are incorrect labels). The last sentence is OK, with ‘equity of access’ being about the service levels (density). The Abstract conclusion needs to be similarly fixed up.

AUTHORS’ RESPONSE: We have revised the ‘Conclusion’ section as per your comments, and we now believe that the conclusion is appropriate. We have also revised the conclusion in the Abstract. Our revision can be found in track changes in the revised manuscript. 

COMMENT: Another nonsensical statement is at lines 257-259 (“Conversely, the lower densities…”), which essentially is saying the ‘lower access (density)’…may explain the ‘lower access (availability)’! Following this, the heading “Variation by access and jurisdiction” needs changing. 

AUTHORS’ RESPONSE: We have revised the stated sentence to make the message being conveyed clearer. We have also revised the heading that reads, “Variation by access and jurisdiction” as “Variation in the density of the three allied health workforce by an objective measure of rurality and jurisdiction”. Our revision can be accessed in track changes in the revised manuscript.

COMMENT: One paragraph didn’t make sense to me – starting at line 189; it twice refers to NSW, SA, Tas, NT, ACT but I can’t determine why they are stated twice.

AUTHORS’ RESPONSE: In the revision, we have grossly revised the stated paragraph to make the message conveyed clearer.

Response to Reviewer #2

Major comments

COMMENT: While this may be the first study of this kind in Australia, this type of study has been done in other countries including USA and Canada. The authors are encouraged to expand introductions and discussions comparing their findings with other studies. Are there any novel findings from this particular study?

AUTHORS’ RESPONSE: We understand that there are a few studies conducted in the USA and Canada regarding the distribution of the allied health workforce, with a focus on physiotherapists and occupational therapists. In the revision, we have added some more new references to expand the ‘Introduction’ and ‘Discussion’ sections as per your comments. Our revision can be found in track changes in the revised manuscript. 

Given that our current study is the first study in Australia that described the distribution of three allied health workforce (physiotherapists, occupational therapists, and podiatrists) according to an objective measure of location/rurality and area-level socioeconomic status, there are several novel findings from this study. These include: (1) In the whole of Australia, the highest densities of the three allied health workforce were found in areas classified as MM 1 and decreased with the increasing levels of the MMM categories. (2) In the whole of Australia, the highest densities of the three allied health workforce were found in areas with IRSAD quintile 5 and decreased with the declining levels of IRSAD quintiles. (3) The patterns of the three allied health workforce densities were unevenly distributed by MM categories and IRSAD quintiles across each state and territory. (4) The densities of these three groups of allied health workforce appeared to be more concentrated in areas classified from MM 1-4 across each state and territory while areas classified from MM 5-7 exhibited less allied health workforce density. 

Minor comments

COMMENT: Abstract: The authors are requested to include the study objective after the background.

AUTHORS’ RESPONSE: In the revision, we have included the study objective in the Abstract. Our revision can be accessed in track changes in the revised manuscript. 

COMMENT: The background needs to describe in more detail why this study is so critical?

AUTHORS’ RESPONSE: In the revision, we have added more descriptions regarding why our current study is very important. Our revision can be found in the track changes in the revised manuscript (see page 8, lines, 128-132).

COMMENT: Discussion section also did not present according to the research purpose and result. The authors are encouraged to Interpret and explain study results, and critically compare to previous studies. For example, at first occupational therapist then physiotherapist and finally podiatrist. 

AUTHORS’ RESPONSE: In the revision, we have added some new explanations and interpretations of the results of our study in the “Discussion” section. We have also re-structured the discussion to align with the study purpose and results. Moreover, we have compared our findings to previous studies conducted in Canada and South Africa. Our revision can be found in the track changes in the revised manuscript. 

COMMENT: Page 17, line 316-317. This sentence is not a limitation as the authors are focusing their objective based on three professions- occupational therapist, physiotherapist and podiatrist. Please drop it.

AUTHORS’ RESPONSE: Thank you for alerting us to this. We have revised it accordingly.

---

## [Decision Letter · Decision Letter 1]

28 Apr 2023

PONE-D-22-18539R1The distribution of registered occupational therapists, physiotherapists, and podiatrists in AustraliaPLOS ONE

Dear Dr. Yisma,

Thank you for submitting your manuscript to PLOS ONE. After careful consideration, we feel that it has merit but does not fully meet PLOS ONE’s publication criteria as it currently stands. Therefore, we invite you to submit a revised version of the manuscript that addresses the points raised during the review process.

We look forward to receiving your revised manuscript.

Kind regards,

Moin Uddin Ahmed

Academic Editor

PLOS ONE

Journal Requirements:

Reviewers' comments:

Reviewer's Responses to Questions

**Comments to the Author**

1. If the authors have adequately addressed your comments raised in a previous round of review and you feel that this manuscript is now acceptable for publication, you may indicate that here to bypass the “Comments to the Author” section, enter your conflict of interest statement in the “Confidential to Editor” section, and submit your "Accept" recommendation.

Reviewer #1: All comments have been addressed

Reviewer #3: (No Response)

2. Is the manuscript technically sound, and do the data support the conclusions?

Reviewer #1: Yes

Reviewer #3: Yes

3. Has the statistical analysis been performed appropriately and rigorously? 

Reviewer #1: Yes

Reviewer #3: Yes

4. Have the authors made all data underlying the findings in their manuscript fully available?

Reviewer #1: Yes

Reviewer #3: Yes

5. Is the manuscript presented in an intelligible fashion and written in standard English?

Reviewer #1: Yes

Reviewer #3: Yes

6. Review Comments to the Author

Reviewer #1: (No Response)

Reviewer #3: This is a descriptive study that presents important information about workforce distribution in Australia. The finding that the allied health workforce is lower per population in more remote regions and those with higher levels of disadvantage is not new, but the detail provided about these three disciplines offers new information. It is important for those involved in workforce planning, development and research to have this information. The paper is also clear, refers to relevant literature and describes the sources of data well. The patterns are described clearly and the implications of these are discussed in the latter sections.

The study would benefit from being more analytical. More detail about specific regions and the patterns and variations of regions would add to this paper. For example, the authors might provide examples of particular places/regions in different states to give more depth. Is it possible to consider how remoteness and disadvantage overlap (and where they do not)?

There are some minor changes for the authors to consider. First, in the results section of the abstract, it states: “However, there was no clear pattern in the distribution of occupational therapists, physiotherapists, and podiatrists density when stratified by the MMM classifications and IRSAD quintiles across each state and territory in Australia.” I could not find evidence of this in the paper; were MM and IRSAD analysed together? Or is this a general sentence, and if so, I would question whether these was not a pattern across MM regions.

Second, I would have liked a sentence in the limitations section to acknowledge variations and differences within states and within these categories. The macro view is important but rural areas and types of disadvantage differ and it is important to acknowledge this. It is easier to recruit a physio to some rural regions than others.

Third, on Line 188, it states “…because SA2 represents communities that interact together socially and economically;” is this correct? Fourth, I was unsure why MM was referred to as an “objective” measure. Finally, there are a couple of editing issues: (1) The phrase on Line 69 “lived shorter” is awkward wording. (2) The sentence on Line 144 “The AHPRA data were extracted in April 2020, which were de-identified” is also awkward wording.

Overall, while a descriptive paper, the evidence of the variation in distribution of the allied health workforce in Australia according to remoteness and levels of disadvantage is important. The detailed graphs presented in this paper would be very useful and I would likely cite this paper.

7. PLOS authors have the option to publish the peer review history of their article (what does this mean?). If published, this will include your full peer review and any attached files.

Reviewer #1: No

Reviewer #3: No

---

## [Author Response · Author response to Decision Letter 1]

7 May 2023

"The distribution of registered occupational therapists, physiotherapists and podiatrists in Australia" (MS: PONE-D-22-18539R1). 

Dear Editor,

We are very grateful for the constructive comments from reviewers. We have taken due consideration of the comments provided and made the necessary revisions to the manuscript. All page and line numbers refer to the marked copy of the manuscript.

Response to Reviewer #3

COMMENT: This is a descriptive study that presents important information about workforce distribution in Australia. The finding that the allied health workforce is lower per population in more remote regions and those with higher levels of disadvantage is not new, but the detail provided about these three disciplines offers new information. It is important for those involved in workforce planning, development and research to have this information. The paper is also clear, refers to relevant literature and describes the sources of data well. The patterns are described clearly and the implications of these are discussed in the latter sections.

AUTHORS’ RESPONSE: Thank you.

COMMENT: The study would benefit from being more analytical. More detail about specific regions and the patterns and variations of regions would add to this paper. For example, the authors might provide examples of particular places/regions in different states to give more depth. Is it possible to consider how remoteness and disadvantage overlap (and where they do not)? 

AUTHORS’ RESPONSE: In the revision, we have added more details regarding the distribution of allied health workforce in specific regions/areas (such as MM 2 regions) across each state and territory in Australia. Our revision can be found in track changes in the revised manuscript (see page 11 and page 13).

Yes, it is possible to consider how remoteness and disadvantage overlap and where they do not. Remoteness and disadvantage can often overlap in areas where access to resources and services is limited due to geographical barriers or economic and social barriers. However, there may be situations where remoteness and disadvantage do not overlap. For instance, individuals living in a remote area may have access to the same resources and services as individuals living in metropolitan area due to availability of infrastructure and technology. Conversely, individuals living in a metropolitan area may experience disadvantage due to poverty, discrimination, or other social and economic factors. In the revision, we have included an explanation highlighting the significance of considering specific context and relevant factors for a thorough analysis of the overlap between remoteness and disadvantage. Our revision can be found tracked changed in the revised manuscript (see page18).

COMMENT: There are some minor changes for the authors to consider. First, in the results section of the abstract, it states: “However, there was no clear pattern in the distribution of occupational therapists, physiotherapists, and podiatrists density when stratified by the MMM classifications and IRSAD quintiles across each state and territory in Australia.” I could not find evidence of this in the paper; were MM and IRSAD analysed together? Or is this a general sentence, and if so, I would question whether these was not a pattern across MM regions.

AUTHORS’ RESPONSE: The MMM and IRSAD data were analyzed independently. In the revision, we have made improvements to the stated sentence to enhance clarity. The revised sentence can be found in the track changes in the “Abstract” section the revised manuscript.

COMMENT: Second, I would have liked a sentence in the limitations section to acknowledge variations and differences within states and within these categories. The macro view is important but rural areas and types of disadvantages differ and it is important to acknowledge this. It is easier to recruit a physio to some rural regions than others. 

AUTHORS’ RESPONSE: Thank you for your feedback. In the revision, we have acknowledged that further studies should explore the significant disparities in the allied health workforce distribution in Australia when classified by MMM categories and IRSAD quintile in detail across rural and disadvantaged areas. This is essential because some rural areas and disadvantaged populations may have unique characteristics that require targeted solutions. Our revision can be found in track changes in the revised manuscript (page 19 and page 20).

COMMENT: Third, on Line 188, it states “…because SA2 represents communities that interact together socially and economically;” is this correct? 

AUTHORS’ RESPONSE: Yes, it is correct. According to the Australian Bureau of Statistics (ABS), SA2s are designed to represents communities that interact together socially and economically. However, we have made some improvements to the wording in order to enhance the clarity of the message being conveyed. Our revision can be found in track changes on page 10.

COMMENT: Fourth, I was unsure why MM was referred to as an “objective” measure. 

AUTHORS’ RESPONSE: The MMM is designed to provide an objective measure of geographic remoteness because it is based on specific criteria and data points that can be quantified and assessed in a consistent manner. However, like any measure, it is important to note that the MMM is not entirely free from subjectivity or limitations. In the revision, for the sake of clarity, we have removed the word "objective" throughout the revised manuscript. Our revision can be found in track changes. 

COMMENT: Finally, there are a couple of editing issues: (1) The phrase on Line 69 “lived shorter” is awkward wording. (2) The sentence on Line 144 “The AHPRA data were extracted in April 2020, which were de-identified” is also awkward wording.

AUTHORS’ RESPONSE: Thank you for the comments. In the revision, we have made some edits to improve the clarity of the conveyed message. Our revision can be found in track changes in the revised manuscript. 

COMMENT: Overall, while a descriptive paper, the evidence of the variation in distribution of the allied health workforce in Australia according to remoteness and levels of disadvantage is important. The detailed graphs presented in this paper would be very useful and I would likely cite this paper.

AUTHORS’ RESPONSE: Thank you.

---

## [Decision Letter · Decision Letter 2]

15 Aug 2023

PONE-D-22-18539R2The distribution of registered occupational therapists, physiotherapists, and podiatrists in AustraliaPLOS ONE

Dear Dr. Yisma,

Thank you for submitting your manuscript to PLOS ONE. After careful consideration, we feel that it has merit but does not fully meet PLOS ONE’s publication criteria as it currently stands. Therefore, we invite you to submit a revised version of the manuscript that addresses the points raised during the review process.

Please accept my apologies for the additional perspectives obtained late in the peer review process. Nevertheless, Reviewer #5 has a few suggestions, appended below, that you may wish to consider to improve your manuscript. In particular, they request additional clarifications about the data sources, which would improve the quality of reporting. Please address this comment:"*"We linked data from three sources: (1) the public registration data of occupational therapists, physiotherapists, and podiatrists obtained from the Australian Health Practitioner Regulation Agency (AHPRA), the Modified Monash Model (MMM) 2019 data from the Australian Department of Health; and the Socio-Economic Indexes for Areas (SEIFA)”- Why are 3 data sources needed? Explain. Do all these agencies have committees in place to study the markers of an effective workforce, to manage demands related to access, and to create and implement public policies aimed at improving their health systems?*"

Regarding the recommendation that you cite specific previously published works, as always we recommend that you please review and evaluate the requested works to determine whether they are relevant and should be cited. It is not a requirement to cite these works.

We would consider that the other suggestions, and the minor points from Reviewer #6, are not required for your manuscript to meet the PLOS ONE publication criteria and may be considered optional. However, if you do decide to make any changes, please ensure that you keep in mind that conclusions must be presented appropriately and should not be overstated, particularly with respect to clinical implications. Please submit your revised manuscript by Sep 29 2023 11:59PM. If you will need more time than this to complete your revisions, please reply to this message or contact the journal office at plosone@plos.org. Please include the following items when submitting your revised manuscript:A rebuttal letter that responds to each point raised by the academic editor and reviewer(s). You should upload this letter as a separate file labeled 'Response to Reviewers'.A marked-up copy of your manuscript that highlights changes made to the original version. You should upload this as a separate file labeled 'Revised Manuscript with Track Changes'.An unmarked version of your revised paper without tracked changes. You should upload this as a separate file labeled 'Manuscript'.If applicable, we recommend that you deposit your laboratory protocols in protocols.io to enhance the reproducibility of your results. Protocols.io assigns your protocol its own identifier (DOI) so that it can be cited independently in the future. For instructions see: https://journals.plos.org/plosone/s/submission-guidelines#loc-laboratory-protocols. Additionally, PLOS ONE offers an option for publishing peer-reviewed Lab Protocol articles, which describe protocols hosted on protocols.io. Read more information on sharing protocols at https://plos.org/protocols?utm_medium=editorial-email&utm_source=authorletters&utm_campaign=protocols.

We look forward to receiving your revised manuscript.

Kind regards,

Marianne Clemence

Staff Editor

PLOS ONE

Journal Requirements:

Reviewers' comments:

Reviewer's Responses to Questions

**Comments to the Author**

1. If the authors have adequately addressed your comments raised in a previous round of review and you feel that this manuscript is now acceptable for publication, you may indicate that here to bypass the “Comments to the Author” section, enter your conflict of interest statement in the “Confidential to Editor” section, and submit your "Accept" recommendation.

Reviewer #4: All comments have been addressed

Reviewer #5: All comments have been addressed

Reviewer #6: All comments have been addressed

2. Is the manuscript technically sound, and do the data support the conclusions?

Reviewer #4: Yes

Reviewer #5: Yes

Reviewer #6: Yes

3. Has the statistical analysis been performed appropriately and rigorously? 

Reviewer #4: Yes

Reviewer #5: Yes

Reviewer #6: Yes

4. Have the authors made all data underlying the findings in their manuscript fully available?

Reviewer #4: Yes

Reviewer #5: Yes

Reviewer #6: Yes

5. Is the manuscript presented in an intelligible fashion and written in standard English?

Reviewer #4: Yes

Reviewer #5: Yes

Reviewer #6: Yes

6. Review Comments to the Author

Reviewer #4: The authors have adequately responded to all comments, and I have no additional concerns.

Reviewer #5: Article: PONE-D-22-18539R2

The distribution of registered occupational therapists, physiotherapists, and podiatrists in Australia

GENERAL COMMENTS

Thank you for allowing me to review this manuscript. The manuscript adhere the PLOS ONE Data Policy. The aim of this study was to describe the national as well as states-and territories-wide distribution of registered occupational therapists, physiotherapists, and podiatrists by measures of rurality and area-level socioeconomic position in Australia. This is a descriptive exploratory quantitative study. It’s an interesting research topic with potential utilization across health disciplines and relevant to the journal. Health care workforce is a global priority to achieve universal health coverage. In my opinion, the paper would need minor changes. Revisions will be necessary. Improve the organization of your paper using the following guidelines.

Those places have all convened committees to study the markers of an effective workforce, to manage the demands related to access, and to create and implement public policies aimed at improving their health care systems.

INTRODUCTION

Abstract: The title of the study and the objective of the study is not matching (include: workforse) Please revise it.

The WHO, rehabilitation group, is being very active in the latter years with activities for strengthening rehab in health systems, with workforce as one of the pillars, with plenty of works, gatherings and citations you could/should use - the most significant. Global Forum on Human Resources for Health, an analysis of the WHO Global Health Observatory Data Repository containing information from 36 countries showed that maintaining a sufficient health care workforce is a global priority and that the effectiveness of that workforce should be determined by calculating the healthcare workforce-to-population ratio. Include these references. It is importantly to systematically search/approach/use/build over the strengths and gaps of the literature on the topic beforehand.

1.Campbell J, Dussault G, Buchan J, Pozo-Martin F, Guerra Arias M, Leone C, Siyam A, Cometto G. A universal truth: No health without a workforce. Forum Report, Third Global Forum on Human Resources for Health, Recife, Brazil. Geneva, Global Health Workforce Alliance and World Health Organization. 2013. http://www.who.int/workforcealliance/knowledge/ resources/hrhreport2013/en/. Accessed 13 Feb 2017.

2. World Health Organization. Dublin Declaration on Human Resources for Health. Fourth Global Forum on Human Resources for Health. 2017. http://www.who.int/hrh/events/Dublin_Declaration-on-HumanResou rces-for-Health.pdf?ua=1.

3. World Health Organization. Global strategy for human resources for health: workforce 2030. Draft for the 69th World Health Assembly 2016. http://www.who.int/hrh/resources/16059_Global_strategyWorkfor ce2030.pdf?ua=1.

I list at least 3 relevant articles (directly focused on the physiotherapy workforce), that were not cited:

Jesus TS, Koh G, Landry M, et al. Finding the "Right-Size" Physical Therapy Workforce: International Perspective Across 4 Countries. Physical therapy 2016;96(10):1597-609. doi: 10.2522/ptj.20160014 [published Online First: 2016/05/07]

Landry MD, Hack LM, Coulson E, et al. Workforce Projections 2010-2020: Annual Supply and Demand Forecasting Models for Physical Therapists Across the United States. Physical therapy 2016;96(1):71-80. doi: 10.2522/ptj.20150010 [published Online First: 2015/10/17]

Shah TI, Milosavljevic S, Trask C, et al. Mapping Physiotherapy Use in Canada in Relation to Physiotherapist Distribution. Physiotherapy Canada Physiotherapie Canada 2019;71(3):213-19. doi: 10.3138/ptc-2018-0023 [published Online First: 2019/11/14]

-What’s new in the scientific literature with this manuscript? Include in introduction.

-The manuscript must be include a hypothesis. Explain the hypothesis.

METHODS

-"We linked data from three sources: (1) the public registration data of occupational therapists, physiotherapists, and podiatrists obtained from the Australian Health Practitioner Regulation Agency (AHPRA), the Modified Monash Model (MMM) 2019 data from the Australian Department of Health; and the Socio-Economic Indexes for Areas (SEIFA)”- Why are 3 data sources needed? Explain. Do all these agencies have committees in place to study the markers of an effective workforce, to manage demands related to access, and to create and implement public policies aimed at improving their health systems?

DISCUSSION

-Include the strengths of the study.

-The final paragraph should leave the reader with your final message within the framework of the hypotheses posed in the Introduction.

- Include the clinical significance of this study over clinicians, patients, and researchers after the study hypothesis.

Reviewer #6: Overall, a very good study synthesizing publicly available data sets to inform workforce planning. Please see a couple of suggestions for your consideration.

You may remove "the latest available data" on Line 79 as it appears redundant.

Also, the text in lines 327-329 appears redundant as the point was made clear in the lines 287-289. However, if it was restated for emphasis, please ignore this suggestion.

7. PLOS authors have the option to publish the peer review history of their article (what does this mean?). If published, this will include your full peer review and any attached files.

Reviewer #4: No

Reviewer #5: **Yes: **Sílvia maria Amado João

Reviewer #6: No

---

## [Author Response · Author response to Decision Letter 2]

5 Sep 2023

"The distribution of registered occupational therapists, physiotherapists and podiatrists in Australia" (MS: PONE-D-22-18539R2). 

Dear Editor,

We are very grateful for the constructive comments from reviewers. We have taken due consideration of the comments provided and made the necessary revisions to the manuscript. All page and line numbers refer to the marked copy of the manuscript.

Response to Reviewer #5 

COMMENT: INTRODUCTION: Abstract: The title of the study and the objective of the study is not matching (include workforce) Please revise it.

AUTHORS’ RESPONSE: In the revision, we have incorporated the term "workforce" into the objective stated in the Abstract. Our revision can be found in track changes in the revised manuscript (Page 2).

COMMENT: The WHO, rehabilitation group, is being very active in the latter years with activities for strengthening rehab in health systems, with workforce as one of the pillars, with plenty of works, gatherings and citations you could/should use - the most significant. Global Forum on Human Resources for Health, an analysis of the WHO Global Health Observatory Data Repository containing information from 36 countries showed that maintaining a sufficient health care workforce is a global priority and that the effectiveness of that workforce should be determined by calculating the healthcare workforce-to-population ratio. Include these references. 

It is importantly to systematically search/approach/use/build over the strengths and gaps of the literature on the topic beforehand. 

1. Campbell J, Dussault G, Buchan J, Pozo-Martin F, Guerra Arias M, Leone C, Siyam A, Cometto G. A universal truth: No health without a workforce. Forum Report, Third Global Forum on Human Resources for Health, Recife, Brazil. Geneva, Global Health Workforce Alliance and World Health Organization. 2013. http://www.who.int/workforcealliance/knowledge/ resources/hrhreport2013/en/. Accessed 13 Feb 2017.

2. World Health Organization. Dublin Declaration on Human Resources for Health. Fourth Global Forum on Human Resources for Health. 2017. http://www.who.int/hrh/events/Dublin_Declaration-on-HumanResou rces-for-Health.pdf?ua=1.

3. World Health Organization. Global strategy for human resources for health: workforce 2030. Draft for the 69th World Health Assembly 2016. http://www.who.int/hrh/resources/16059_Global_strategyWorkfor ce2030.pdf?ua=1.

I list at least 3 relevant articles (directly focused on the physiotherapy workforce), that were not cited:

Jesus TS, Koh G, Landry M, et al. Finding the "Right-Size" Physical Therapy Workforce: International Perspective Across 4 Countries. Physical therapy 2016;96(10):1597-609. doi: 10.2522/ptj.20160014 [published Online First: 2016/05/07]

Landry MD, Hack LM, Coulson E, et al. Workforce Projections 2010-2020: Annual Supply and Demand Forecasting Models for Physical Therapists Across the United States. Physical therapy 2016;96(1):71-80. doi: 10.2522/ptj.20150010 [published Online First: 2015/10/17]

Shah TI, Milosavljevic S, Trask C, et al. Mapping Physiotherapy Use in Canada in Relation to Physiotherapist Distribution. Physiotherapy Canada Physiotherapie Canada 2019;71(3):213-19. doi: 10.3138/ptc-2018-0023 [published Online First: 2019/11/14]

AUTHORS’ RESPONSE: Thank you for suggesting these references. In the revised manuscript, we have incorporated citations for most of the mentioned references (see Page 4). 

COMMENT: What’s new in the scientific literature with this manuscript? Include in introduction.

AUTHORS’ RESPONSE: While previous studies have explored aspects of health workforce distribution, few have comprehensively analysed the allied health workforce's density on a national and state/territory level, considering both socioeconomic characteristics and geographical location. Our manuscript uniquely aimed that combining the measures of location/rurality and area-level socioeconomic position as well as the total resident population to provide a new insight into the complex interplay between health workforce availability, socioeconomic factors, and geographic disparities. This approach is important to deliver policy relevant insights aimed at promoting equitable access to allied health workforce. In the revision, we have incorporated what our manuscript adds to the scientific literature in the Introduction section. Our revision can be found in track changes in the revised manuscript (see Page 7).

COMMENT: The manuscript must be include a hypothesis. Explain the hypothesis.

AUTHORS’ RESPONSE: Given that our study is a descriptive quantitative study, our primary focus is on summarizing and presenting data based on our research objective/question rather than testing a specific hypothesis. However, we have provided/added a clear description of our research objective that can serve as a guiding framework for our study (see Introduction page 7).

COMMENT: METHODS: "We linked data from three sources: (1) the public registration data of occupational therapists, physiotherapists, and podiatrists obtained from the Australian Health Practitioner Regulation Agency (AHPRA), the Modified Monash Model (MMM) 2019 data from the Australian Department of Health; and the Socio-Economic Indexes for Areas (SEIFA)”- Why are 3 data sources needed? Explain. Do all these agencies have committees in place to study the markers of an effective workforce, to manage demands related to access, and to create and implement public policies aimed at improving their health systems?

AUTHORS’ RESPONSE: We used the three data sources (by linking each of them) to explore the interplay between allied health workforce availability, measure of location/rurality, and socioeconomic position. This approach is important to deliver policy relevant insights aimed at promoting equitable healthcare access and outcomes. The first data source, the AHPRA, plays a foundational role by providing information about registered professionals. This dataset includes details about occupational therapists, physiotherapists, and podiatrists, providing valuable insights into their numbers and locations. The second source, the MMM 2019 data, offers a geographic classification system that categorises areas based on their remoteness and population size. The third source, the SEIFA, delves into socio-economic factors at different geographic levels, offering insights into the relative socio-economic advantage or disadvantage of different areas. By incorporating SEIFA data, we gain a deeper understanding of the socio-economic context within which healthcare practitioners operate. 

Regarding the presence of committees within these agencies for studying workforce markers, managing access demands, and creating/implementing public policies: 

• The AHPRA is primarily responsible for the registration and regulation of health practitioners. While AHPRA itself might not have committees specifically focused on studying workforce markers or creating public policies, it collaborates with various professional boards (e.g., Physiotherapy Board, Occupational Therapy Board) to ensure practitioner standards and regulatory measures are in place. 

• The Australian Department of Health, which provides the Modified Monash Model (MMM) data, may have committees or departments dedicated to healthcare workforce planning, policy development, and access management. These committees might analyze data trends to ensure that healthcare services are appropriately distributed, especially in underserved areas. 

• SEIFA is a product of the ABS. The ABS might not have specific committees focused on healthcare, but the SEIFA data is widely used by various government agencies and researchers to understand socio-economic disparities and inform policy decisions, including those related to healthcare. 

Therefore, while these agencies may not have dedicated committees solely for healthcare workforce markers, access demands, and policy creation, they are likely involved in these areas through collaboration, data provision, and policy implementation within their broader mandates. In the revision, we provide a clear description of why these data sources are need in our study. Our revision can be found in track changes (see Page 8.) 

COMMENT: DISCUSSION: Include the strengths of the study.

AUTHORS’ RESPONSE: We have already included the strength of our study in the 'Discussion' section (see page 19).

COMMENT: The final paragraph should leave the reader with your final message within the framework of the hypotheses posed in the Introduction.

AUTHORS’ RESPONSE: In the Conclusion section, we have highlighted the notable disparity in the distribution of registered occupational therapists, physiotherapists, and podiatrists when categorised by measures of location/rurality and area-level socioeconomic status across Australian jurisdictions. These findings align with the research question/objective outlined in our Introduction, which focused on describing the density of these three groups of allied health workforce based on location/rurality and area-level socioeconomic status across Australian jurisdictions. The pronounced geographic imbalances in access to allied health workforce underscore the need for targeted policy interventions and comprehensive workforce planning. These measures are crucial in ensuring that healthcare services remain accessible and equitable for all Australians, irrespective of their geographic location or socioeconomic status. In the revision, we have added some more explanation in the last paragraph to enhance the clarity of our final message. Our revision can be found in track changes (see Page 21). 

COMMENT: Include the clinical significance of this study over clinicians, patients, and researchers after the study hypothesis.

AUTHORS’ RESPONSE: In the revision, we have added some discussion regarding the significance of our findings. Our revision can be accessed track changed in the ‘Discussion’ section in the revised manuscript (see Page 19).

Response to Reviewer #6

COMMENT: You may remove "the latest available data" on Line 79 as it appears redundant. 

AUTHORS’ RESPONSE: Thank you. We have revised it as per the suggestion. 

COMMENT: Also, the text in lines 327-329 appears redundant as the point was made clear in the lines 287-289. However, if it was restated for emphasis, please ignore this suggestion.

AUTHORS’ RESPONSE: Thank you for pointing this. We have restated the stated text for emphasis and explanation.

---

## [Editor Report · Decision Letter 3]

10 Sep 2023

The distribution of registered occupational therapists, physiotherapists, and podiatrists in Australia

PONE-D-22-18539R3

Dear Dr. Yisma,

We’re pleased to inform you that your manuscript has been judged scientifically suitable for publication and will be formally accepted for publication once it meets all outstanding technical requirements.

Kind regards,

Marianne Clemence

Staff Editor

PLOS ONE
---

## [Editor Report · Acceptance letter]

13 Sep 2023

PONE-D-22-18539R3 

The distribution of registered occupational therapists, physiotherapists, and podiatrists in Australia 

Dear Dr. Yisma:

I'm pleased to inform you that your manuscript has been deemed suitable for publication in PLOS ONE. Congratulations! Your manuscript is now with our production department. 

Kind regards, 

on behalf of

Dr Marianne Clemence 

Staff Editor

PLOS ONE